# SELF-EVOLVING LANGUAGE MODELS VIA SIMPLE GENERATOR-VERIFIER GAMES

## ABSTRACT

Post-training language models often depends on costly external signals such as human annotations or domain-specific rewards. As an alternative, we explore model self-evolution through the lens of simple generator–verifier games. A single base model plays both roles—generating candidate solutions and verifying/improving their quality—to construct preference data for fine-tuning. To extract reliable signals from noisy self-verification, we leverage *thresholded majority voting*, which approximates high-precision preference pairs. The approach enables self-evolution on synthetic logical reasoning and realistic mathematical reasoning tasks, even when models initially perform poorly. For example, on the *Knights and Knaves* benchmark, accuracy rises from 31.0% to **40.7%** with single-turn verification, **42.2%** with multi-turn verification, **44.1%** with iterative training, and **44.8%** with curriculum learning. Notably, models trained only on easier instances generalize effectively to harder test data, demonstrating *emergent easy-to-hard generalization*. These results show that simple generator-verifier games can unexpectedly enhance reasoning in small models, offering a new perspective on concurrent research in self-improvement and RL with verifiable rewards.

## 1 INTRODUCTION

Large language models (LLMs) have made strong progress on complex reasoning tasks (Comanici et al., 2025; Yang et al., 2025b; DeepSeek-AI et al., 2025). A central driver of this progress has been post-training techniques that refine model outputs using feedback signals. Paradigms such as Supervised Finetuning (SFT), Reinforcement Learning from Human Feedback (RLHF), and Reinforcement Learning from Verifiable Rewards (RLVR) (Lambert et al., 2024; Gao et al., 2024; Wang et al., 2025) have become standard for improving performance in downstream reasoning tasks.

However, these approaches remain constrained by their reliance on external supervision. Human annotation is costly, slow, and limited by domain expertise (Bassi et al., 2025; Giorgi et al., 2025; Plank, 2022). Verifiable reward signals, such as code execution or exact-match math answers, are confined to narrow domains (Liu et al., 2025; Wu et al., 2025). Current methods overlook a vast landscape of tasks where external ground truth is unavailable, ambiguous, or impractical to obtain.

This situation raises a fundamental question: *Can a single language model self-improve without external supervision?* This deep question has led to a wave of recent research on model self-evolution. Approaches like test-time reinforcement learning (TTRL) (Zuo et al., 2025) and R-Zero (Huang et al., 2025) have demonstrated the power of consensus-based signals, such as majority voting, in tasks where structured outputs admit straightforward comparison. Similarly, Absolute Zero (Zhao et al., 2025a) effectively utilizes external environments, leveraging code execution as an objective verifier for domains with executable semantics. Meanwhile, methods such as INTUITOR (Zhao et al., 2025b), LSP (Kuba et al., 2025), and EMPO (Zhang et al., 2025) have trained models without external data by using online reinforcement learning. These efforts inspire a systematic analysis of the core principles of self-evolution. Specifically, we are motivated to study: 1) whether we can directly use the same model as both generator and verifier to bootstrap self-evolution, and 2) how to best construct a reinforcement learning dataset in an offline fashion. If this leads to improvements, it would be a general approach that can be widely applied to downstream domains with minimal assumptions on reward verifiability, environment executability, or output structures, and it would effectively reduce the computational burden of training with online reward signals.

Figure 1: Illustration of the generator–verifier games. We use the same base model as a generator $G$ and a verifier $V$. The generator outputs several responses, and the verifier $V$ labels them as correct or incorrect. In the single-turn *SimpleGV* (left), we run the verifier multiple times and assign labels based on a threshold. In the multi-turn *RevisionGV* (right), we use the last two responses if they switch from incorrect to correct according to the verifier. We train with the positive/negative samples via offline preference optimization.

In this work, we study several types of self-evolution through generator–verifier (GV) games. At a high level, a single base model is instantiated in two roles: a *generator*, which proposes candidate solutions, and a *verifier*, which evaluates their quality. In the simplest *single-turn game*, the verifier forms preference pairs $(y_{\text{win}}, y_{\text{lose}})$ by labeling candidate responses (verifier-as-a-judge). Since we cannot always expect the verifier to be better than the generator, we also explore thresholded voting to aggregate multiple verifier responses. In a richer *multi-turn game*, the verifier iteratively provides feedback and the generator revises its outputs, producing higher-quality alternatives. We also explore extensions of these variants, where we use iterative training or curriculum learning.

A central challenge is that self-verification is noisy, as models may mislabel solutions. To address this, we use a *thresholded majority voting* method. The verifier is queried multiple times per candidate. Define the *correctness rate* as the fraction of times the verifier says a response is correct. We label a response as *positive* if its correctness rate exceeds a threshold $\tau$, and as *negative* if its correctness rate less than $(1 - \tau)$, and we discard it otherwise. This filters out ambiguous cases and yields "confident" preference pairs, extracting a reliable signal from imperfect self-assessment.

We validate this framework on both synthetic and realistic reasoning tasks. For example, on the *Knights and Knaves* (KK) logical reasoning benchmark, accuracy improves from 31.0% for the base model to **40.7%** with single-turn verification, **42.2%** with multi-turn verification, **44.1%** with iterative training, and **44.8%** with curriculum learning. Similar improvements are observed across diverse mathematical reasoning benchmarks, including GSM8K, MATH, and TabMWP, leading to performance competitive with previous self-evolution methods. These results show that even smaller models, which initially perform poorly, can substantially enhance their reasoning abilities through simple generator–verifier games, achieving performance nearly on par with supervised methods. Beyond absolute accuracy gains, self-evolution also enables strong *easy-to-hard generalization*: models trained only on simpler KK instances (2–3 people) transfer to harder ones (4–8 people), where the KK problem complexity and solution space grow sharply as we add more people.

Our detailed analysis provides the following new contributions and insights:

- **Simple yet General Framework for Self-Evolution:** We map out and study various generator–verifier games. We focus on when a single model, without external labels or environments, both generates and evaluates its own outputs to produce preference data. Despite this simplicity, such a framework can improve performance across multiple real and synthetic reasoning benchmarks.

- **Principles for Self-Evolution:** Next, we identify methods that consistently improve performance. These include (i) enhancing the reliability of verifier feedback through a thresholded majority voting scheme and (ii) using the generator and verifier in a multi-turn fashion, where the model revises responses rather than just labeling them as correct or incorrect.

- **Bootstrapping and Generalization:** Going further, we also show that iterative refinement and curriculum learning can additionally enhance self-evolution, and that training on easier cases transfers effectively to harder ones, demonstrating impressive easy-to-hard generalization.

## 2 PRELIMINARIES

**Generator–Verifier Game.** A single base model $\mathcal{M}$ is instantiated in two roles using different system prompts: a *generator* $\mathcal{G}$ and a *verifier* $\mathcal{V}$.[1] Given an unlabeled prompt set $\mathcal{D}$ and a base model $\mathcal{M}$, we define a generator–verifier game

$$\mathsf{GV}(\mathcal{M}, \mathcal{D}, T) \rightarrow \mathcal{P},$$

which instantiates $\mathcal{M}$ as generator $\mathcal{G}$ and verifier $\mathcal{V}$, and runs for $T$ rounds. For a query $q \in \mathcal{D}$, the generator produces $k$ candidates

$$\hat{Y}(q) = \{\hat{y}_1, \ldots, \hat{y}_k\}, \quad \hat{y}_i \sim \mathcal{G}(\cdot \mid q),$$

while the verifier assigns binary judgments $\mathcal{V}(q, \hat{y}_i) \in \{\texttt{Correct}, \texttt{Incorrect}\}$. Then, from these interactions we extract preference pairs

$$(y_w, y_l) \in \mathcal{P} \quad \text{iff} \quad \mathcal{V}(q, y_w) = \texttt{Correct}, \;\; \mathcal{V}(q, y_l) = \texttt{Incorrect}.$$

**Single-turn vs. Multi-turn Verification.** In the *single-turn* case, $T = 1$ and preference pairs are obtained directly from static judgments. In the *multi-turn* case ($T > 1$), the generator refines its outputs based on verifier feedback:

$$\hat{y}^{(t+1)} \sim \mathcal{G}(\cdot \mid q, f(\mathcal{V}(q, \hat{y}^{(t)}))),$$

where $f : \{\texttt{Correct}, \texttt{Incorrect}\} \rightarrow \mathcal{X}_{\text{feedback}}$ maps verifier judgments into textual feedback prompts. A pair is extracted whenever

$$\mathcal{V}(q, \hat{y}^{(t)}) = \texttt{Incorrect}, \;\; \mathcal{V}(q, \hat{y}^{(t+1)}) = \texttt{Correct}.$$

**Preference Learning.** The generator–verifier game yields a dataset of preference triples $\mathcal{D}_{\text{pref}} = \{(x, y_w, y_l)\}$, where $x$ is a prompt, $y_w$ is a preferred response, and $y_l$ is a dispreferred one. Preference learning fine-tunes a policy $\pi_\theta$ so that preferred responses are assigned higher probability than dispreferred ones. We apply *Direct Preference Optimization* (DPO) (Rafailov et al., 2023) which refines $\pi_\theta$ relative to a fixed reference policy $\pi_{\text{ref}}$ by minimizing

$$\mathcal{L}_{\text{DPO}}(\pi_\theta; \pi_{\text{ref}}) = -\mathbb{E}_{(x,y_w,y_l) \sim \mathcal{D}_{\text{pref}}} \left[ \log \sigma \left( \beta \log \frac{\pi_\theta(y_w|x)}{\pi_{\text{ref}}(y_w|x)} - \beta \log \frac{\pi_\theta(y_l|x)}{\pi_{\text{ref}}(y_l|x)} \right) \right],$$

where $\beta > 0$ is a parameter controlling the sharpness of preference alignment. Intuitively, this loss increases the relative likelihood of $y_w$ over $y_l$ while keeping $\pi_\theta$ close to reference policy $\pi_{\text{ref}}$, ensuring both preference alignment and stability during fine-tuning.

**Iterative Preference Learning.** We may apply GV repeatedly. Starting from $\mathcal{M}_0 = \mathcal{M}$, define

$$\mathcal{P}_t = \mathsf{GV}(\mathcal{M}_{t-1}, \mathcal{D}_t, T), \quad \mathcal{M}_t = \texttt{Finetune}(\mathcal{M}_{t-1}, \mathcal{P}_t).$$

This yields a sequence $\{\mathcal{M}_t\}_{t=1}^{T}$ that progressively refines reasoning ability. Unlike online RL, all updates are offline since $\mathcal{P}_t$ is fixed once generated.

**Curriculum Learning.** If prompts can be partitioned by difficulty, $\mathcal{D} = \mathcal{D}_{\text{easy}} \cup \mathcal{D}_{\text{hard}}$, we first generate $\mathcal{P}_{\text{easy}} = \mathsf{GV}(\mathcal{M}, \mathcal{D}_{\text{easy}}, T)$ and fine-tune on it, before proceeding to $\mathcal{P}_{\text{hard}}$.

### 2.1 EXPERIMENT SETUP

**Models.** We use the `gemma-3-it` (Gemma Team, 2025) and `Qwen-2.5-Instruct` (Yang et al., 2025a) families as base models. Since the same model is instantiated as both generator and verifier, we employ instruction-tuned variants rather than raw base models.

**Datasets.** Our controlled experiments use *Knights-and-Knaves* (KK) (Xie et al., 2024), a synthetic dataset for reasoning. Each instance describes a group of inhabitants who are either *knights* (always truthful) or *knaves* (always lying). The task is to infer each inhabitant's identity from their statements. The difficulty scales with the number of people, as the search space grows exponentially and demands deeper logical inference. This structured setting provides a testbed for isolating the effects of self-evolution. To assess generality, we also evaluate on four reasoning benchmarks:

---

[1]We provide the exact prompts in Appendix C.

- *GSM8K* (Cobbe et al., 2021): grade-school math word problems requiring multi-step arithmetic.
- *MATH500*: a medium-scale subset of the MATH benchmark (Hendrycks et al., 2021) spanning diverse levels of difficulty.
- *MATHHard*: the hardest subset of MATH (difficulty level 5), with advanced problem-solving.
- *TabMWP* (Lu et al., 2022): math word problems involving structured tabular data.

For training, we **only use unlabeled prompts**, without access to ground-truth solutions. Specifically, we use the KK training set for logical reasoning tasks and OpenThoughts3 (Guha et al., 2025) for mathematical reasoning tasks. Notably, OpenThoughts3 includes problems that are not directly verifiable (e.g., proofs and scientific question answering), highlighting the importance of a general and self-contained verifier that can analyze free-form outputs.

**Evaluation Protocol.** We report exact-match accuracy, requiring the model's output to perfectly match the ground-truth solution. This strict metric eliminates ambiguity from partial matches. For each query, we generate one sample using temperature 0.7 and average results over four random seeds. The same evaluation protocol is applied consistently across KK and the other benchmarks.

## 3 SIMPLEGV: SINGLE-TURN VERIFICATION WITH VERIFIER-AS-A-JUDGE

We begin with the simplest setting, where a single model serves as both generator and verifier, and the verifier directly judges the quality of generated responses without iterative feedback. This "verifier-as-a-judge" setup constitutes the minimal generator–verifier game, allowing us to isolate its effectiveness before introducing further refinements. It tests our core hypothesis: that a model's latent ability to evaluate a solution, even if imperfect, can be harnessed to improve its own generation ability. We illustrate this SimpleGV approach in Figure 1. We implicitly assume that a model's ability to *verify* a candidate is, on average, more reliable than its ability to *generate* one from scratch. We view SimpleGV as distilling these latent verification capabilities into a usable training signal.

### 3.1 THRESHOLDED MAJORITY VOTING FOR MORE ACCURATE VERIFICATION

A central challenge is the noisiness of an unsupervised verifier. Smaller models in particular may mislabel solutions or produce inconsistent judgments, contaminating the preference dataset. To mitigate this, we use a **thresholded majority voting** method. For each candidate $\hat{y}$, the verifier is queried $n$ times, producing binary judgments $Z_j = \mathbf{1}\{\mathcal{V}^{(j)}(q, \hat{y}) = \texttt{Correct}\}$. We then compute the empirical correctness rate $\hat{p}(q, \hat{y}) = \frac{1}{n}\sum_{j=1}^{n} Z_j$. A candidate is labeled $\texttt{Positive}$ if $\hat{p}(q, \hat{y}) \geq \tau$, $\texttt{Negative}$ if $1 - \hat{p}(q, \hat{y}) \geq \tau$, and discarded otherwise. Note that thresholding at 0.5 falls back to regular majority voting. This procedure filters out ambiguous cases and yields high-confidence preference pairs, extracting a reliable signal from noisy self-assessment. As shown in Figure 2, increasing the threshold effectively improves verification accuracy.

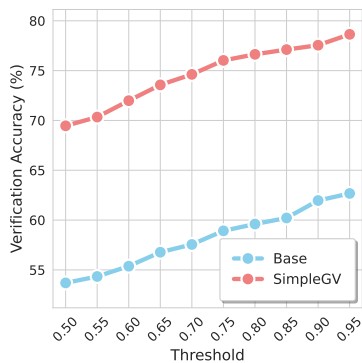

Figure 2: Verification accuracy on the KK training set for `gemma-3-4b-it` and its SimpleGV variant under different thresholds.

Table 1 summarizes results on five reasoning benchmarks. For baseline methods, we evaluate their released models on the corresponding benchmarks, and also refer to their origianl report. SimpleGV consistently improves over base models without requiring ground-truth labels, supervised signals, or external environments. Unlike prior methods that depend on executable environments or online reinforcement learning, SimpleGV operates directly on free-form text using offline optimization, yet still achieves substantial gains. Moreover, not only does generation improve, but verification accuracy also increases (Figure 2), demonstrating a process of *co-evolution* where both roles reinforce one another.

Table 1: Results on five reasoning benchmarks. For SimpleGV, we train with 20K samples obtained from OpenThoughts3. We compare baselines: INUITOR, two Absolute Zero (AZR) models, and GRPO. In the RL type column, we list whether it uses online or offline training. The supervised (Supervis.) column shows if the method uses additional labels or reward models; the environment (Environ.) column shows if it uses external tools (* from original report).

| Model/Algorithm | RL Type | Supervis. | Environ. | Benchmarks | | | | |
| --- | --- | --- | --- | --- | --- | --- | --- | --- |
| | | | | GSM8K | MATH500 | MATHHard | TabMWP | KK |
| *Gemma 3* | | | | | | | | |
| gemma-3-4b-it | / | / | / | **89.2*** | 75.8 (0.4) | 53.7 (0.2) | 84.5 (0.2) | 31.0 (1.3) |
| **SimpleGV (ours)** | Offline | No | No | 89.0 (0.1) | **77.4** (0.6) | **55.1** (0.4) | **87.4** (0.3) | **33.2** (0.5) |
| gemma-3-12b-it | / | / | / | 94.4* | 85.6 (0.1) | 69.1 (0.3) | 95.2 (0.2) | 47.5 (0.7) |
| *Qwen 2.5* | | | | | | | | |
| Qwen2.5-7B-Instruct | / | / | / | 90.2 (0.4) | 73.5 (0.5) | 49.7 (0.3) | 91.9 (0.2) | **18.1** (0.9) |
| Base + INTUITOR | Online | No | No | 87.3* | 75* | / | / | / |
| Base + AZR | Online | No | Yes | 84.0 (0.4) | 74.4* | 32.8 (0.5) | 68.8 (0.7) | 5.1 (0.4) |
| Base + AZR-Coder | Online | No | Yes | 83.4 (0.1) | 72.6* | 40.1 (0.7) | 78.5 (0.5) | 8.5 (0.4) |
| Base + GRPO | Online | Yes | No | 82.9* | 75* | / | / | / |
| **SimpleGV (ours)** | Offline | No | No | **90.6** (0.1) | **76.0** (0.7) | **51.5** (0.4) | **92.3** (0.2) | 17.6 (0.5) |
| Qwen2.5-14B-Instruct | / | / | / | 94.8* | 77.1 (0.5) | 54.5 (0.3) | 93.7 (0.3) | 26.4 (0.3) |

## 3.2 SELF-IMPROVEMENT VERSUS MODEL SIZES

We next examine how SimpleGV scales with model size. Figure 3 reports results on `gemma-3-it` for 1B, 4B, and 12B, with 27B included as an approximate upper bound. All models are trained on KK instances with 2–3 people and evaluated on test sets spanning 2–8 people.

We find that self-improvement occurs at all scales but manifests differently. For smaller models (1B), verifier judgments are noisy and improvements modest. Medium-scale models (4B and 12B), however, achieve substantial gains, showing that the generator–verifier framework becomes increasingly effective as model capacity grows. While the 27B model establishes a performance roofline, the 12B model with SimpleGV approaches this level, indicating that self-evolution enables weaker models to close much of the gap to stronger baselines.

## 3.3 SELF-IMPROVEMENT VERSUS DATA SIZES

A natural question is how the amount of self-generated data influences downstream performance. To investigate this, we experiment with the `gemma-4b-it` model, varying the size of the preference dataset constructed from OpenThoughts3 using the generator–verifier game. We consider datasets of 5K, 10K, 20K, and 40K preference pairs (by using a comparable number of initial questions), while keeping all hyperparameters fixed.

As shown in Figure 4, enlarging the preference set yields clear gains at small–moderate scales (e.g., 5k → 20k), but improvements taper thereafter and can even regress at 40k for TabMWP and KK. This reflects *diminishing returns* from simply adding more self-generated pairs: beyond a moderate size, redundancy and verifier noise begin to dominate, suggesting that tighter filtering and greater prompt diversity are more effective than sheer volume. We note a small dip at 5k samples on GSM8K and KK; we attribute this to small-sample variance and mild prompt-distribution skew in early batches, which diminishes as the dataset grows in size and diversity.

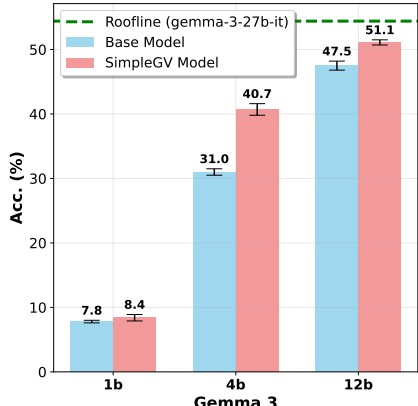

Figure 3: Effect of model size on SimpleGV performance. Models are trained on KK instances with 2–3 people and evaluated on 2–8 people.

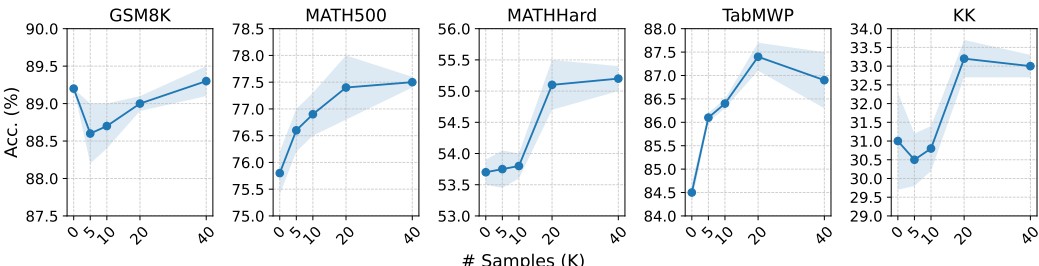

Figure 4: Effect of data size on SimpleGV performance. Models are trained on OpenThoughts3 dataset. Accuracy improves across benchmarks as the number of training samples increases. For TabMWP and KK, performance slightly degrades when data increases from 20k to 40k. Subplots show mean accuracy (4 runs) with shaded standard-error regions.

## 3.4 ITERATIVE PREFERENCE LEARNING

We next test whether repeating the preference-learning loop yields further gains. As shown in Table 2, performance improves across iterations, though gains diminish over time. The first iteration provides the largest boost, while subsequent ones yield smaller increments. Importantly, training only on easier KK instances (2–3 people) improves generalization to harder ones (4–8 people): three rounds of unsupervised DPO raise accuracy from 31.0% to 44.1%, approaching the 46.6% obtained with a supervised verifier. This highlights that iterative preference learning not only compounds improvements but also supports strong *easy-to-hard generalization*.

Table 2: Iterative DPO results on KK. Accuracy (%) is averaged over subsets (2–3, 4–5, and 6–8 people) with standard deviations in parentheses. Rows compare different verifier thresholds $\tau$. Oracle results (in gray) use ground-truth labels for verification. Three rounds of unsupervised DPO improve accuracy from 31.0% to 44.1%, approaching the 46.6% achieved with an oracle verifier.

| model | 2–3 ppl. | 4–5 ppl. | 6–8 ppl. | All |
|---|---|---|---|---|
| gemma-3-4b-it | 62.0 (1.7) | 31.0 (0.9) | 10.3 (1.3) | 31.0 (1.3) |
| SimpleGV, $\tau$=0.6 | 70.9 (1.9) | 45.4 (3.8) | 17.5 (2.9) | **40.7** (2.8) |
| Oracle Verifier | 78.4 (1.8) | 52.6 (2.1) | 21.4 (1.4) | 46.6 (1.7) |
| SimpleGV, $\tau$=0.6 → SimpleGV, $\tau$=0.5 | 69.5 (1.9) | 44.8 (2.7) | 18.1 (1.3) | 40.4 (1.9) |
| → SimpleGV, $\tau$=0.6 | 74.2 (2.1) | 46.9 (2.5) | 20.3 (0.8) | **43.3** (1.6) |
| → SimpleGV, $\tau$=0.7 | 71.5 (1.8) | 46.8 (1.9) | 20.8 (2.0) | 42.7 (1.9) |
| → SimpleGV, $\tau$=0.8 | 72.2 (1.6) | 48.1 (2.2) | 18.6 (1.3) | 42.4 (1.7) |
| → Oracle Verifier | 82.4 (0.8) | 58.6 (2.3) | 30.2 (2.5) | 53.2 (1.9) |
| SimpleGV, $\tau$=0.6 → SimpleGV, $\tau$=0.6 → SimpleGV, $\tau$=0.5 | 75.2 (1.6) | 49.6 (2.0) | 19.7 (2.0) | **44.1** (1.9) |
| → SimpleGV, $\tau$=0.6 | 74.5 (1.4) | 46.0 (1.8) | 18.8 (1.7) | 42.5 (1.7) |
| → SimpleGV, $\tau$=0.7 | 70.8 (1.6) | 46.3 (2.4) | 16.3 (1.1) | 40.4 (1.6) |
| → SimpleGV, $\tau$=0.8 | 72.2 (2.6) | 45.9 (2.4) | 20.7 (1.7) | 42.6 (2.2) |
| → Oracle Verifier | 85.0 (1.1) | 61.9 (1.8) | 25.0 (2.4) | 52.6 (1.9) |

## 3.5 CURRICULUM LEARNING

We also study how scheduling problem difficulty impacts self-evolution. In *curriculum learning*, we first train on easier problems before progressing to harder ones. This contrasts with a *random mixing* baseline that uses both easy and hard problems jointly from the start.

As shown in Table 3, curriculum learning consistently outperforms random mixing. Starting with simpler problems reduces verifier noise and provides more reliable supervision in early stages, enabling more stable self-evolution. Moreover, curriculum learning improves easy-to-hard transfer: training on KK with 2–3 people and then 4–5 people yields an average accuracy of 44.8%, compared to 31.0% for the base model and 41.2% for random mixing. This demonstrates that staged progression not only stabilizes training but also enhances *easy-to-hard generalization*.

Table 3: Curriculum learning results on KK. Accuracy (%) is averaged over subsets (2–3, 4–5, and 6–8 people) with standard deviations in parentheses. Rows compare different verifier thresholds $\tau$. Oracle results (in gray) use ground-truth labels for verification. Curriculum learning outperforms random mixing baselines and enables easy-to-hard generalization.

| model | 2–3 ppl. | 4–5 ppl. | 6–8 ppl. | All |
|---|---|---|---|---|
| gemma-3-4b-it | 62.0 (1.7) | 31.1 (0.9) | 10.3 (1.3) | 31.0 (1.3) |
| KK2345 w/ SimpleGV, $\tau$=0.5 | 68.6 (1.7) | 44.3 (1.6) | 17.6 (1.5) | 39.8 (1.6) |
| SimpleGV, $\tau$=0.6 | 67.2 (1.5) | 39.9 (2.1) | 14.7 (1.5) | 36.9 (1.6) |
| SimpleGV, $\tau$=0.7 | 71.0 (1.8) | 42.9 (1.1) | 15.4 (1.1) | 39.1 (1.3) |
| SimpleGV, $\tau$=0.8 | 72.9 (2.5) | 46.1 (1.9) | 16.7 (1.8) | **41.1 (2.0)** |
| Oracle Verifier | 80.9 (1.6) | 54.4 (1.9) | 23.8 (1.5) | 48.8 (1.6) |
| KK23 w/ SimpleGV, $\tau$=0.6 | 70.9 (1.9) | 45.4 (3.8) | 17.5 (2.9) | 40.7 (2.8) |
| KK23 w/ SimpleGV, $\tau$=0.6 → KK45 w/ SimpleGV, $\tau$=0.5 | 74.1 (1.4) | 49.9 (1.8) | 19.4 (1.4) | 43.7 (1.5) |
| → KK45 w/ SimpleGV, $\tau$=0.6 | 76.2 (2.0) | 49.7 (1.8) | 20.6 (2.1) | **44.8 (2.0)** |
| → KK45 w/ SimpleGV, $\tau$=0.7 | 72.4 (2.0) | 48.6 (3.5) | 20.3 (2.1) | 43.2 (2.5) |
| → KK45 w/ SimpleGV, $\tau$=0.8 | 68.4 (1.9) | 44.3 (2.0) | 18.6 (1.5) | 40.1 (1.7) |
| → KK45 w/ Oracle Verifier | 80.8 (1.2) | 60.9 (1.6) | 29.8 (2.9) | 53.3 (2.0) |
| KK23 w/ Oracle Verifier | 78.4 (1.8) | 52.6 (2.1) | 21.4 (1.4) | 46.6 (1.7) |
| KK23 w/ Oracle Verifier → KK45 w/ SimpleGV, $\tau$=0.5 | 80.3 (1.7) | 53.7 (2.5) | 22.9 (2.4) | 48.0 (2.2) |
| → KK45 w/ SimpleGV, $\tau$=0.6 | 77.7 (1.2) | 56.2 (1.6) | 21.6 (2.2) | 47.5 (1.7) |
| → KK45 w/ SimpleGV, $\tau$=0.7 | 76.3 (1.5) | 53.9 (1.9) | 19.2 (1.7) | 45.4 (1.7) |
| → KK45 w/ SimpleGV, $\tau$=0.8 | 78.7 (2.0) | 51.8 (2.2) | 19.8 (1.9) | 45.7 (2.0) |
| → KK45 w/ Oracle Verifier | 84.2 (1.6) | 60.2 (2.0) | 28.2 (1.7) | 53.3 (1.8) |

## 3.6 COST ANALYSIS

Finally, we analyze the computational trade-offs of the generator–verifier framework. Self-evolution requires both multiple candidate generations and multiple verifier passes. The total cost thus depends on the number of generations per query ($n_1$) and verifier passes per candidate ($n_2$). We vary $n_1$ and $n_2$ systematically across thresholds from 0.5 to 0.8 and report average performance in Figure 5.

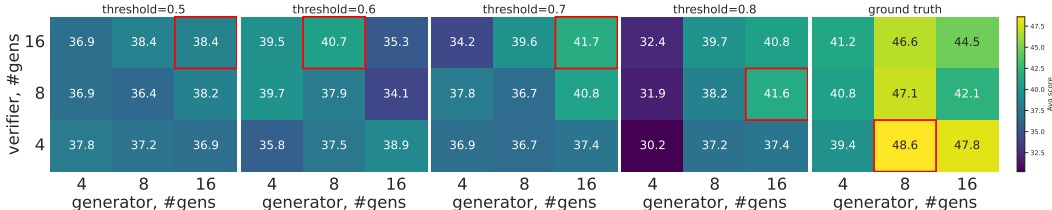

Figure 5: Cost–performance trade-offs in SimpleGV. Grids show average accuracy with generations $n_1$ (x-axis) and verifier passes $n_2$ (y-axis) across thresholds 0.5–0.8; the right-most plot uses oracle verification (ground-truth labels). Accuracy improves as $n_1$ and $n_2$ increase, though very high thresholds (e.g., 0.8) cause data sparsity.

For `gemma-3-4b-it`, threshold $\tau = 0.7$ achieves the best balance of precision and recall, reaching an average accuracy of 41.7%. Performance scales with both $n_1$ and $n_2$, though costs grow linearly. These results highlight a practical trade-off: larger generator and verifier budgets yield higher accuracy, but moderate configurations already achieve strong results at much lower cost. As a rule of thumb, we conclude that scaling up verifier computation is typically more cost-effective than scaling up generator computation; however this may depend on the specific task and dataset.

## 4 REVISIONGV: MULTI-TURN GENERATOR–VERIFIER GAME

While single-turn verification demonstrates the feasibility of using a verifier-as-a-judge, it does not fully exploit the base model's ability to provide feedback and analyze solutions. For example, there are cases where an initial solution may be partially correct but contain errors. In these cases, the verifier model can identify these errors, going beyond just labeling the solution as incorrect. This in turn enables the generator and verifier to interact across multiple rounds. Specifically, the generator can revise its output in response to feedback, and it can progressively improve the solution.

Table 4: Results on KK for 1B, 4B, and 12B models. Accuracy (%) is averaged over subsets (2–3, 4–5, 6–8 people) with standard deviations in parentheses. Rows compare SimpleGV at different verifier thresholds $\tau$, RevisionGV, and an oracle verifier (gray) using ground-truth labels.

| model | 2–3 ppl. | 4–5 ppl. | 6–8 ppl. | All |
|---|---|---|---|---|
| gemma-3-1b-it | 20.9 (1.1) | 4.9 (1.1) | 1.0 (0.4) | 7.8 (0.2) |
| SimpleGV, $\tau$=0.5 | 15.2 (1.6) | 3.5 (0.6) | 0.8 (0.3) | 5.7 (0.6) |
| SimpleGV, $\tau$=0.6 | 17.0 (1.8) | 2.0 (0.8) | 0.3 (0.2) | 5.6 (0.1) |
| SimpleGV, $\tau$=0.7 | 19.0 (1.7) | 3.3 (0.8) | 0.4 (0.3) | 6.5 (0.3) |
| SimpleGV, $\tau$=0.8 | 23.8 (2.6) | 4.5 (1.1) | 0.8 (0.4) | **8.4** (0.5) |
| Oracle Verifier | 32.6 (2.2) | 10.0 (0.9) | 0.7 (0.4) | 12.5 (0.4) |
| RevisionGV | 22.4 (2.4) | 4.7 (0.8) | 0.2 (0.2) | 7.8 (0.5) |
| gemma-3-4b-it | 62.0 (1.7) | 31.0 (0.9) | 10.3 (1.3) | 31.0 (0.5) |
| SimpleGV, $\tau$=0.5 | 70.8 (1.2) | 39.1 (2.6) | 16.3 (1.7) | 38.4 (0.7) |
| SimpleGV, $\tau$=0.6 | 70.9 (1.9) | 45.4 (3.8) | 17.5 (2.9) | 40.7 (0.9) |
| SimpleGV, $\tau$=0.7 | 70.1 (1.6) | 43.9 (1.0) | 16.4 (1.9) | 39.6 (0.6) |
| SimpleGV, $\tau$=0.8 | 70.4 (1.6) | 44.6 (2.1) | 16.0 (1.1) | 39.7 (0.5) |
| Oracle Verifier | 78.4 (1.8) | 52.7 (2.1) | 21.4 (1.4) | 46.6 (0.7) |
| RevisionGV | 75.8 (3.0) | 46.4 (2.6) | 17.1 (1.5) | **42.2** (0.4) |
| gemma-3-12b-it | 77.7 (1.7) | 51.9 (2.3) | 24.4 (1.2) | 47.5 (0.7) |
| SimpleGV, $\tau$=0.5 | 78.3 (1.1) | 53.7 (1.5) | 24.0 (1.7) | 48.0 (0.5) |
| SimpleGV, $\tau$=0.6 | 84.8 (1.8) | 55.0 (1.3) | 26.2 (1.3) | 51.1 (0.4) |
| SimpleGV, $\tau$=0.7 | 83.0 (0.9) | 56.3 (0.6) | 24.0 (1.1) | 50.1 (0.2) |
| SimpleGV, $\tau$=0.8 | 80.5 (2.1) | 53.9 (2.5) | 21.2 (2.1) | 47.5 (1.0) |
| Oracle Verifier | 86.8 (1.8) | 60.3 (1.4) | 27.0 (2.0) | 53.6 (0.5) |
| RevisionGV | 84.8 (1.0) | 58.7 (2.6) | 27.5 (1.1) | **52.8** (1.0) |

We refer to this setup as **RevisionGV**, or *multi-turn generator–verifier verification*. RevisionGV enables iterative correction: the verifier provides feedback, and the generator revises its outputs in subsequent rounds. As detailed in Section 2 and Figure 1, the RevisionGV process generates a preference pair when the generator revises an incorrect solution into a correct one based on the verifier's feedback. In other words, RevisionGV is not just a multi-turn game, but the method is also a test of the model's ability to perform in-context learning from its own critiques.

**Results.** We evaluate RevisionGV on the KK benchmark using `gemma-3-it` (1B, 4B, and 12B) as the base model, and compare it against SimpleGV. As shown in Table 4, RevisionGV consistently outperforms SimpleGV across all thresholds and all difficulty levels. RevisionGV on `gemma-3-12b-it` achieves an average accuracy of up to 52.8%, approaching the performance of oracle ground-truth filtering (53.6%). This underscores the strength of self-evolved preference data and demonstrates that the model can not only identify its own errors but also *actively correct them based on self-feedback*—a more sophisticated form of self-improvement than passive selection. Results in Table 4 also reveal a scaling trend. For the 1B model, SimpleGV is better than RevisionGV. On the other hand, for the 4B and 12B, we see a consistent improvement with RevisionGV.

**Discussion.** Our findings from RevisionGV suggest that as model capacity grows, its dual roles as generator and verifier become increasingly effective. Intuitively, this is possible because the verifier feedback is more detailed, and also the generator can better incorporate this feedback when revising the solution. It is not clear if this trend will continue or saturate with even larger models, which is an area of future work. Finally, we note that RevisionGV takes advantage of the offline nature of our training, where we can use natural language feedback to create better preference data.

## 5 RELATED WORK

Recent work has explored ways to improve model performance without explicit supervision. Zhao et al. (2025a) introduce "Absolute Zero", a method for generating coding problems and verifiable

solutions. Huang et al. (2025) present the R-Zero method, with improvements over Absolute Zero, using the same general methodology. However, the reliance on verification through majority voting limits general applicability. Zuo et al. (2025) propose confidence as an unsupervised reward to enhance performance. Yet, this approach necessitates tasks that support meaningful majority voting. Zweiger et al. (2025), with Self-Adapting Language Models, employ an external validation set during inference-time optimization, departing from self-evolution. Similarly, SPC Critic requires seed data from a larger LLM (Chen et al., 2025). Wen et al. (2025) introduce Internal Coherence Maximization, a new self-scoring based finetuning algorithm, which is orthogonal to our approach.

Another area is self-refinement, where models use their own feedback to enhance generated text (Madaan et al., 2023). Further advancements include training models to explicitly self-correct their reasoning steps via reinforcement learning (Kumar et al., 2024) and employing search-based algorithms to rectify logical chains (Kim et al., 2025). The quality of the training data is also important and raises challenges for generating high-quality synthetic prompts (Yu et al., 2025).

A trend for better reasoning is the study of methods that minimize the need for custom-trained rewards. Going beyond standard RL (Zhao et al., 2025c; Ji et al., 2024), this area includes reinforced self-play (Zhao et al., 2025a), synthetic code edits (Piterbarg et al., 2025), co-evolutionary collective feedback (Yuan et al., 2025), self-logits evolution (Zhang et al., 2024), and DPO extensions (Tu et al., 2025). To generate robust rewards, researchers have explored using confident reasoning traces (Jang et al., 2025), external feedback models (Sun et al., 2023), teaching reward models to "think" (Zhou et al., 2025), autorating RAG contexts (Joren et al., 2025), entropy-based methods (Zhang et al., 2025), and ways to avoid spurious rewards (Shao et al., 2025). These ideas extend to adapting models during deployment, such as test-time training for distribution shifts (Sun et al., 2020) and TTRL (Zuo et al., 2025), and other self-adaptation ideas (Zweiger et al., 2025).

## 6 CONCLUSION

We studied a self-evolution framework, where a single language model acts as both generator and verifier to improve reasoning without external supervision. We showed that the model can produce reliable pairs for preference tuning. Our experiments across reasoning benchmarks with free-form outputs demonstrated that self-evolution yields consistent improvements, getting close to supervised baselines. Our work revealed three key takeaways: (i) scaling self-generated data enhances performance; (ii) larger models provide more reliable self-judgments; and (iii) multi-turn verification with RevisionGV outperforms voting-based SimpleGV. Expanding on (iii), we saw that feedback-driven corrections in RevisionGV provide a stronger learning signal than simply discriminating between correct and incorrect solutions with SimpleGV. Ultimately, we provided new evidence that external signals like human labels or domain-specific rewards is *not a prerequisite* for improving models.

For future work, it would be interesting to use complementary functionalities of the single base model, beyond critiquing or judging solutions. Another avenue is to explore how self-generated data affects training dynamics; it would be a fundamental insight to theoretically analyze the limits or stability of optimizing with self-generated data, using only input questions as the human-generated part. Finally, it is an open direction to find methods that work for very small models ($\leq$1B) or very large models, where we posit that we need to utilize a dataset that is not too challenging and not too easy, as a pre-requisite to effectively self-evolve. In particular, if a model is over-saturated on some particular task, it would be interesting to see how much further self-improvement is possible.

**Limitations.** The SimpleGV and RevisionGV generator–verifier games require multiple generations and verifier passes, making it computationally intensive. On the other hand, when we consider small, open source LLMs, we may be willing to pay the trade-off in cost vs. accuracy. Another aspect to consider is that performance is also sensitive to thresholds, which currently need minor task-specific tuning. That being said, a threshold between 0.6 and 0.7 seems reliable for multiple downstream tasks, meaning the optimization is fairly robust. Nonetheless, addressing efficiency and adaptive calibration are promising directions for future work. Finally, the self-evolution process is fundamentally limited by the base model's latent knowledge. While it can effectively surface and refine existing reasoning abilities, it is not designed to discover knowledge or reasoning strategies that are entirely outside its initial training distribution. The self-evolution process amplifies what the model knows, but might struggle to teach it what it does not know at all.

ETHICS STATEMENT

This work studies self-evolving language models in a controlled research setting using only publicly available datasets (e.g., synthetic logic puzzles and benchmark reasoning tasks). No human subjects or personally identifiable information were involved in data collection. Our methods do not rely on sensitive or private data, and we make no claims beyond the intended research scope. While language models have the potential for misuse, our study focuses exclusively on understanding their self-improvement dynamics under safe, synthetic conditions. We highlight that broader deployment of such models should carefully consider issues of bias, fairness, and responsible use. This research adheres to the ICLR Code of Ethics and complies with principles of transparency, integrity, and reproducibility.

REPRODUCIBILITY STATEMENT

We have taken several steps to ensure the reproducibility of our results. All datasets used in this study (synthetic Knights and Knaves problems and publicly available reasoning benchmarks) are clearly described in the main text and appendix, with details of experimental setup provided. We include complete descriptions of the generator–verifier training protocols, preference learning objectives, and evaluation metrics in the paper. Hyperparameters, model sizes, and training schedules are documented in the appendix.

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

## A  LLM USAGE

We used LLMs solely as a writing assistant to polish the paper. Specifically, LLMs were employed to improve grammar, clarity, and flow of exposition, without contributing to the research ideas, experimental design, analyses, or results. All scientific content, methodology, and conclusions presented in this work were conceived and authored entirely by the listed authors.

## B  EXPERIMENT DETAILS

### B.1  TRAINING AND EVALUATION SETTINGS

We consider two main training settings:

- **Synthetic reasoning.** Models are trained on the *Knights and Knaves* (KK) training set (restricted to instances with 2–3 people) and evaluated on the held-out KK test set covering 2–8 people.

- **Mathematical reasoning.** Models are trained on the *OpenThoughts3* dataset and evaluated on four benchmarks: GSM8K, MATH500, MATHHard, TabMWP, as well as the KK test set.

No additional preprocessing was applied beyond the original dataset splits.

### B.2  MODELS AND OPTIMIZATION

We use instruction-tuned `gemma-3-it` models (1B, 4B, 12B) and `Qwen-2.5-7B-Instruct`. All models are fully fine-tuned (no parameter-efficient adaptation). Optimization uses AdamW with a sequence length of 4096 and batch size of 256. Training schedules are as follows:

- **Gemma-1B:** learning rate $7.5 \times 10^{-7}$, 3 epochs.
- **Gemma-4B:** learning rate $5.0 \times 10^{-7}$, 3 epochs.
- **Gemma-12B:** learning rate $2.5 \times 10^{-7}$, 3 epochs.
- **Qwen-7B:** learning rate $7.5 \times 10^{-7}$, 5 epochs.

### B.3  GENERATOR–VERIFIER SETUP

Unless otherwise specified, we use $n_1 = 8$ candidate generations per query and $n_2 = 16$ verifier passes per candidate. We set the confidence threshold to $\tau = 0.6$. For RevisionGV (multi-turn verification), the generator revises responses for up to 4 rounds, with the verifier providing free-form feedback that ends with a structured label.

### B.4  ITERATIVE AND CURRICULUM LEARNING

For iterative preference learning, we repeat the generator–verifier loop for 2–3 rounds. To isolate the effect of iteration, we reuse the same prompt set at each round rather than re-sampling. For curriculum learning, difficulty levels are determined by the KK dataset (based on the number of people). By default, models are trained on KK with 2–3 people before being evaluated on harder cases.

## C  PROMPTS

We provide the prompts we use for the specific KK verifier, the generic verifier, and the generic reviser. In Appendix C.1, we perform experiments showing how the verifier performance changes with prompts and model sizes. This complements our results in Figure 2 on verifier accuracy (for the specific KK prompt) as the model trains. For the main experiments, we use "Generic Prompt 3" for our OpenThoughts data collection, as it performs best on 2 of 3 datasets (Musique & KK).

**Knights and Knaves (KK) Specific Prompt for Verifier**

```
CRITIC_POS_SYMBOL = "CRITIC RESULT: SOLUTION IS CORRECT"
CRITIC_NEG_SYMBOL = "CRITIC RESULT: SOLUTION IS INCORRECT"

CRITIC_SYSTEM_MESSAGE = f"""You are a critic tasked with analyzing
a solution to a logical reasoning problem and determining whether
the solution correctly deduces the identities of characters
(e.g., knights or knaves). Carefully examine whether the
explanation uses valid deductive logic, correctly interprets
the statements, and exhaustively considers all cases. Pay
attention to whether contradictions are correctly identified
and if the conclusion logically follows from the analysis.

- If the solution is logically sound and complete, finalize
your critique with '{CRITIC_POS_SYMBOL}'.

- If the solution contains reasoning flaws, invalid
assumptions, missed cases, or unsupported conclusions, explain
these issues in detail and finalize with '{CRITIC_NEG_SYMBOL}'."""

CRITIC_PROMPT_TEMPLATE = (
    f"{CRITIC_SYSTEM_MESSAGE}\n\n"
    "## Problem\n{query}\n\n"
    "## Solution\n{response}\n\n"
    "## Your response"
)
```

**Generic Prompt for Verifier**

```
CRITIC_POS_SYMBOL = "CRITIC RESULT: SOLUTION IS CORRECT"
CRITIC_NEG_SYMBOL = "CRITIC RESULT: SOLUTION IS INCORRECT"

CRITIC_SYSTEM_MESSAGE = f"""You are a meticulous and critical
logic expert specializing in math, puzzles, logic and
factuality problems. Your task is to analyze a proposed
solution to the problem below and determine if it is correct.

To do this, go through the problem and then go through each
step in the answer very carefully, checking if there are any
inconsistencies or contradictions with the conditions in the
problem. End your response with '{CRITIC_POS_SYMBOL}' if the
solution has no contradictions/ inconsistencies to the
conditions in the question. Otherwise, end your response with
'{CRITIC_NEG_SYMBOL}' if the solution has one more
contradictions or inconsistencies."""

CRITIC_PROMPT_TEMPLATE = (
    f"{CRITIC_SYSTEM_MESSAGE}\n\n"
    "## Problem\n{query}\n\n"
    "## Solution\n{response}\n\n"
    "## Your response"
)
```

**Generic Revision Prompt**

```
ANALYSIS_SYMBOL = "ANALYSIS:"
REVISOR_ANSWER_SYMBOL = "REVISED SOLUTION:"

REVISOR_SYSTEM_MESSAGE = f"""You are given a logical reasoning
problem, an initial solution, and a critic's feedback on that
solution. Your task is to revise the original solution so that
it is correct, logically sound, and fully aligned with the
problem's requirements. Your revision should strictly follow
the instructions in the problem and address all the issues
raised by the critic.

Your response should be in the following format:
{ANALYSIS_SYMBOL} ...
{REVISOR_ANSWER_SYMBOL} ..."""

REVISOR_PROMPT_TEMPLATE = (
    f"{REVISOR_SYSTEM_MESSAGE}\n\n"
    "## Problem\n{query}\n\n"
    "## Solution\n{response}\n\n"
    "## Critic's Feedback\n{critic_feedback}\n\n"
    "## Your response"
)
```

## C.1 VERIFIER RESULTS

We perform a deep dive here in the accuracy of the unsupervised verifier. For different datasets, we compare an unsupervised labeled (correct/incorrect) versus using a strong model (Gemini 2.5 Pro) that has access to the ground truth as the true label. We then compute the agreement with the supervised strong model as a measure of unsupervised verifier accuracy. Furthermore, we compare using three generic prompts (e.g., not specific to reasoning) against a dataset-specific prompt. We present our results in Tables 5, 6, 7, and 8.

Table 5: Gemma 4B, MATH, Verifier Accuracy. We sample 60 question and compare precision and accuracy (correct/incorrect labels) of an unsupervised model (no ground truth) versus supervised Gemini 2.5 Pro (with ground truth answers). For the "Single" cases we average over 3 runs, standard deviation in parens. We also evaluate taking the Majority (Maj.) over 3 samples.

| Prompt Type | Prec. Single | Acc. Single | Prec. Maj. | Acc. Maj. |
|---|---|---|---|---|
| Specific Prompt MATH | 85.1 (0.0) | 85.0 (0.0) | 85.1 | 85.0 |
| Generic Prompt 1 | **89.0 (0.1)** | **88.9 (0.8)** | **88.9** | **88.3** |
| Generic Prompt 2 | 87.0 (1.8) | 86.7 (2.7) | 87.0 | 86.7 |
| Generic Prompt 3 | 81.4 (1.4) | 80.6 (1.6) | 83.0 | 81.7 |

Table 6: Gemma 4B, KK, Verifier Accuracy. We sample 60 question and compare precision and accuracy (correct/incorrect labels) of an unsupervised model (no ground truth) versus supervised Gemini 2.5 Pro (with ground truth answers). For the "Single" cases we average over 3 runs, standard deviation in parens. We also evaluate taking the Majority (Maj.) over 3 samples.

| Prompt Type | Prec. Single | Acc. Single | Prec. Maj. | Acc. Maj. |
|---|---|---|---|---|
| Specific Prompt KK | 45.2 (4.1) | 53.3 (1.4) | 50.0 | 55.0 |
| Generic Prompt 1 | 52.3 (1.8) | 57.8 (2.1) | 51.3 | 56.7 |
| Generic Prompt 2 | 52.8 (2.2) | 58.3 (2.7) | 52.9 | 58.3 |
| Generic Prompt 3 | **55.1 (0.6)** | **60.0 (0.0)** | **55.2** | **60.0** |

Table 7: Gemma 27B, MATH, Verifier Accuracy. We sample 60 question and compare precision and accuracy (correct/incorrect labels) of an unsupervised model (no ground truth) versus supervised Gemini 2.5 Pro (with ground truth answers). For the "Single" cases we average over 3 runs, standard deviation in parens. We also evaluate taking the Majority (Maj.) over 3 samples.

| Prompt Type | Prec. Single | Acc. Single | Prec. Maj. | Acc. Maj. |
|---|---|---|---|---|
| Specific Prompt MATH | 94.0 (0.7) | 91.7 (0.0) | 94.6 | 91.7 |
| Generic Prompt 1 | 94.2 (0.8) | 94.4 (0.8) | 94.7 | 95.0 |
| Generic Prompt 2 | **92.5 (0.7)** | **92.2 (0.8)** | **91.5** | **91.7** |
| Generic Prompt 3 | 91.9 (0.8) | 90.0 (1.4) | 91.4 | 90.0 |

Table 8: Gemma 27B, KK, Verifier Accuracy. We sample 60 question and compare precision and accuracy (correct/incorrect labels) of an unsupervised model (no ground truth) versus supervised Gemini 2.5 Pro (with ground truth answers). For the "Single" cases we average over 3 runs, standard deviation in parens. We also evaluate taking the Majority (Maj.) over 3 samples.

| Prompt Type | Prec. Single | Acc. Single | Prec. Maj. | Acc. Maj. |
|---|---|---|---|---|
| Specific Prompt KK | 84.7 (3.2) | 68.3 (2.7) | 85.7 | 70.0 |
| Generic Prompt 1 | **90.5 (2.0)** | **90.0 (3.6)** | **91.3** | **91.7** |
| Generic Prompt 2 | 77.6 (0.8) | 75.0 (1.4) | 76.9 | 75.0 |
| Generic Prompt 3 | 80.8 (3.6) | 72.8 (4.2) | 80.0 | 73.3 |

