# OpenReview forum: "Self-Evolving Language Models via Simple Generator-Verifier Games"
_ICLR.cc/2026/Conference — Submitted to ICLR 2026_

### Official Review · Reviewer_b2eD · 2025-10-21

**Soundness:** 2
**Presentation:** 2
**Contribution:** 1
**Rating:** 2
**Confidence:** 5

**Summary:**

This paper proposes a self evolution framework in which a single language model acts as both generator and verifier to construct preference data for self improvement. The method introduces two variants, SimpleGV (single turn verification) and RevisionGV (multi turn verification), and applies iterative DPO. The authors evaluate the approach mainly on mathematical reasoning benchmarks.

**Strengths:**

The paper is clearly written and conceptually easy to follow.

**Weaknesses:**

**Limited novelty.**\
The proposed framework of self-generation and self-verification is not new. Similar iterative preference learning or self-rewarding approaches have already been explored in prior works such as [1,2,3,4,5]. The paper differs only in minor details (e.g., thresholded verification, curriculum scheduling), which do not constitute meaningful methodological novelty. It also overlaps heavily with [6] in both structure and training procedure.

**Lack of proper comparison and discussion.**\
The paper does not compare against the most relevant prior works on self-improvement or self-rewarding LMs. There is also little discussion on how the proposed method differs conceptually or empirically from existing iterative preference optimization frameworks.

**Weak experimental validation.**\
The experiments are narrow in scope (e.g., training on a single dataset, using only two models, and evaluating mostly on math relaative easy reasoning tasks). There are no experiments on other domains (e.g., coding) or stronger reasoning benchmarks (e.g., AIME, AMC). As a result, the claims of general self-evolution remain unsubstantiated.

**Cost inefficiency.**\
The generator–verifier framework requires multiple generations and verification passes, increasing computational cost significantly. However, the observed performance gains are modest, suggesting poor cost–performance trade-off.

**Overall.** The method appears to be a minor extension of existing self-reward/self-evolution frameworks, without demonstrating clear advantages or generality.

Reference\
[1] Self-Rewarding Language Models\
[2] Self-consistency preference optimization\
[3] Meta-Rewarding Language Models: Self-Improving Alignment with LLM-as-a-Meta-Judge\
[4] CREAM: Consistency Regularized Self-Rewarding Language Models\
[5] ARIES: Stimulating Self-Refinement of Large Language Models by Iterative Preference Optimization\
[6] ReVISE: Learning to Refine at Test-Time via Intrinsic Self-Verification

**Questions:**

See the weakness above.

---

### Official Review · Reviewer_Lso3 · 2025-10-31

**Soundness:** 2
**Presentation:** 3
**Contribution:** 2
**Rating:** 2
**Confidence:** 4

**Summary:**

The paper studies generator-verifier games where the same instruction‑tuned model acts as a generator that proposes multiple answers and a verifier that judges whether each answer is correct. Preference pairs are then formed and used to fine‑tune the model with DPO. The paper studies two variants: SimpleGV, which involves single‑turn, verifier as a judge with thresholded majority voting, and RevisionGV, which uses multiple turns, generator revises responses using verifier feedback. The method is evaluated on synthetic KK puzzles and math benchmarks, with strong KK improvements.

**Strengths:**

* Timely problem: Overall the problem the paper aims to solve is good. Reducing dependence on human labels or domain-specific verifiers is very important, and a single model generator/verifier is very a simple formulation.
* Clear Writing: The paper does well at explaining how they form preference pairs and provides concrete settings. I found this easy to follow.
* Ablations: The ablations on KK were really useful and important giving a sense of how the synthetic task behaves.

**Weaknesses:**

I think this paper needs additional evaluations to be stronger and seems unfinished. I list my concerns below:

* Improvements on Real Datasets: The paper finds really small improvements on datasets like GSM-8K or MATH-500 in comparison to the synthetic task. I think this needs more analysis, see one concern below on this thread.
* Distribution Leakage: I was not familiar with OpenThoughts3, but I looked into it and found that it was really close in distribution to GSM-8K and MATH-500. It aggregated reasoning problems. I wonder if this leads to inflated gains and the modest gains seen are from using OpenThoughts3 rather than the set up seen here.
* Hyperparameters: How are hyperparameters selected? My reading right now is that they are selected using the testing set...
* Verifier Signal: The core claim of the paper is that we can extract reliable signals from noisy self-verification via thresholded voting. But, when measured against a strong model with access to ground truth, the unsupervised verifier is only ~60% accurate on KK with Gemma‑4B. At scale, this is far from real preference labels. Even when making the model larger, you're still topping out at around 91%.
* Small Models: In the abstract, the authors claim that the approach "enhances reasoning in small models" yet Gemma‑1B shows little to no benefit. In the limitations section, this is acknowledged as well. I would reduce claims on generality in the abstract implied upfront.
* Novelty: Conceptually, I am having trouble differentiating between this method and self-refinement. To me, the paper is in this way, purely empirical. The major problem with this is that the main improvement the paper demonstrates is with KK and very limited improvement on the math benchmarks. Without stronger gains or deeper analysis, the contribution doesn't feel very strong to me.,

**Questions:**

* Could the authors discuss OpenThoughts3 and its training distribution a bit more?

---

### Official Review · Reviewer_pUW8 · 2025-10-31

**Soundness:** 3
**Presentation:** 2
**Contribution:** 2
**Rating:** 2
**Confidence:** 3

**Summary:**

The paper proposes Simple Generator–Verifier (GV) Games, a self-evolution framework that enables a single language model to generate and verify its own outputs without external supervision. By constructing high-confidence preference pairs through thresholded majority voting and optimizing via DPO, the model self-improves its reasoning ability. The authors demonstrate consistent gains in reasoning accuracy and easy-to-hard generalization across multiple benchmarks, showing that simple self-verification can approach supervised performance efficiently.

**Strengths:**

1. The paper conducts extensive experiments across multiple benchmarks, model scales, and ablation settings, demonstrating robustness and reproducibility.

2. The problem setting—learning from unlabeled prompts and unverifiable tasks—is timely and important.

3. The idea of turning a model’s own verification capability into a generator–verifier game, rather than relying solely on majority voting, is novel.

**Weaknesses:**

1. The paper lacks sufficient explanation of baseline methods such as INTUITOR, Absolute Zero (AZR), and AZR-Coder—it is unclear how these baselines are implemented or differ from the proposed approach.

2. Although the related work section mentions Absolute Zero and R-Zero, there is no direct empirical or conceptual comparison, making it difficult to assess the advantage of the proposed method.

3. Without a comparison to an external or supervised verifier, it is hard to evaluate how much of the improvement comes from self-verification itself versus incidental effects of DPO fine-tuning.

4. Different tables use different model backbones (e.g., Table 1 uses Qwen2.5-7B-Instruct, while Table 4 uses Gemma-3-1B-it), raising concerns about consistency and fairness in comparison.

5. Figure 2 shows that the SimpleGV verifier accuracy improves over the base model, but since the verifier is not explicitly trained, it is unclear why or how this improvement emerges.

6. In Table 1, several baselines perform worse than their original reports (e.g., GRPO with Qwen2.5-7B drops from 90.2 → 82.9), yet no interpretation or justification is provided.
 ---
Presentation.\
a. The gray-highlighted rows in the tables are difficult to interpret .\
b. The numbers in Figure 5 are too small and hard to distinguish.\
c. The texts in Table 3 are overly long, making it hard to understand each configuration clearly

**Questions:**

1. How would performance change if the verifier were trained or fine-tuned separately rather than sharing parameters with the generator?

2. Is the model update applied only to the generator role, or does it indirectly improve verification ability as well?

3. Could the authors clarify whether iterative GV training leads to verifier drift (i.e., changes in its reliability over iterations)?

4. Could the authors provide more explanation and evidence for lines 264–266 — specifically, what causes “redundancy and verifier noise to begin to dominate” beyond a moderate model size?

5. Could the authors include an error case analysis to better illustrate the failure modes of self-evolution?

---

### Official Review · Reviewer_eTrK · 2025-11-02

**Soundness:** 2
**Presentation:** 2
**Contribution:** 2
**Rating:** 2
**Confidence:** 5

**Summary:**

The paper proposes as test time inference method relying on majority voting with an LLM as a judge utilized for a single-turn and multi-turn setup.

**Strengths:**

- The paper evaluates on several empirical domains such as knights and knaves and also performs ablations over thresholds and data and model sizes

**Weaknesses:**

- LLM as a judge and Majority Voting/Ensembling has been well studied in prior work (e.g [1]) thus is unclear the main contributions of this work with respect to these approaches
- Several baselines (e.g MCTS) from the test time inference literature hasn't been compared to in this work, making it hard to judge the contribution of the proposed method

[1] Scaling LLM Test-Time Compute Optimally can be More Effective than Scaling Model Parameters

**Questions:**

See weaknesses above.

---

### Meta-Review · Area_Chair_xr6A · 2026-01-08

**Summary:**

Reviewers consistently viewed the core idea (single model as generator + verifier, thresholded majority voting to build preference pairs for DPO/iterative self-training) not sufficiently novel relative to prior LLM-as-judge, majority-voting/ensembling, and self-reward literature. They also questioned whether the empirical evidence supports the paper’s broader claims: gains are strong on the synthetic Knights-and-Knaves setting but modest on real math benchmarks; evaluation scope is narrow, and cost/performance tradeoffs may be unfavorable due to multiple generations + verification passes. Several reviewers additionally flagged missing key baselines. The reviewers are consistent to give a rejection score and there is no author rebuttal, thus I recommend reject.

**Reviewer Concerns:**

There is no author rebuttal

**Reviewer Scores:**

There is no author rebuttal

---

### Decision · Program_Chairs · 2026-01-26

Reject